# Melting temperatures of MgO under high pressure by micro-texture analysis

T. Kimura[1,2], H. Ohfuji[1], M. Nishi[1,3] & T. Irifune[1,3]

Periclase (MgO) is the second most abundant mineral after bridgmanite in the Earth's lower mantle, and its melting behaviour under pressure is important to constrain rheological properties and melting behaviours of the lower mantle materials. Significant discrepancies exist between the melting temperatures of MgO determined by laser-heated diamond anvil cell (LHDAC) and those based on dynamic compressions and theoretical predictions. Here we show the melting temperatures in earlier LHDAC experiments are underestimated due to misjudgment of melting, based on micro-texture observations of the quenched samples. The high melting temperatures of MgO suggest that the subducted cold slabs should have higher viscosities than previously thought, suggesting that the inter-connecting textural feature of MgO would not play important roles for the slab stagnation in the lower mantle. The present results also predict that the ultra-deep magmas produced in the lower mantle are peridotitic, which are stabilized near the core–mantle boundary.

[1] Geodynamics Research Center, Ehime University, Ehime 790-8577, Japan. [2] Department of Earth and Planetary Materials Science, Graduate School of Science, Tohoku University, Sendai 980-8578, Japan. [3] Earth-Life Science Institute, Tokyo Institute of Technology, Tokyo 152-8550, Japan. Correspondence and requests for materials should be addressed to T.K. (email: tomoaki.kimura.c1@tohoku.ac.jp).

Investigating the melting behaviours of terrestrial materials at high pressure is important in addressing generation of magmas and associated differentiation processes in the Earth's deep interior[1]. Experiments using multianvil apparatus have provided detailed melting relations of model mantle materials at high pressure and temperature, which have been used to constrain the formation of the deep magma ocean and subsequent chemical differentiation of the deep mantle[2,3]. However, these experiments have been limited to pressures up to $\sim 30$ GPa, equivalent to the depths of the uppermost region of the lower mantle, due to the limitation in pressure and temperature generation in conventional multianvil technology using tungsten carbide as the second-stage anvils. Although recent developments of this technology using sintered polycrystalline diamond anvils expanded the pressure limit to $\sim 100$ GPa[4], the temperatures available under such pressures have been limited to below 2,000 K, which are insufficient to study the melting relations of mantle materials.

Static high-pressure experiments using laser-heated diamond anvil cell (LHDAC)[5–9], as well as those by dynamic compression[10–12] and ab initio calculations[13–17], have been attempted to constrain the melting relations of peridotitic mantle rocks and of relevant major minerals, such as $MgSiO_3$ bridgmanite and MgO, under the lower mantle conditions. The composition of the partial melt in the lower mantle can be reasonably modelled by the eutectic composition in the melting relation of the $MgO–MgSiO_3$ system[18], which can be evaluated by the melting curves of these two end-member minerals. However, the melting relation of this binary system has not been well constrained because of the large discrepancies in experimentally and theoretically determined their melting curves, particularly for MgO (ref. 19). The melting behaviour of the MgO end-member is highly controversial among previous studies: a pioneering work using LHDAC reported a melting curve with a $dT_m/dP$ of $\sim 30$ K GPa$^{-1}$ at zero pressure, which was extrapolated from the experiments up to $\sim 30$ GPa[6], while several theoretical computations gave substantially higher $dT_m/dP$ of 90–120 K GPa$^{-1}$ (refs 13–15). A recent shock experiment showed the melting of the high-pressure B2 phase of MgO at $\sim 14,000$ K at $\sim 650$ GPa[12], which is consistent with the theoretically predicted higher melting slopes rather than the result of the LHDAC experiment. Furthermore, the melting curves of MgO inferred from recent two melting experiments on the (Mg,Fe)O solid solution[20,21] do not agree with each other. Such inconsistency of the earlier melting curves of MgO is a major reason for the poorly constrained melting relations of the lower mantle materials[19].

Determination of melting temperatures of MgO under the lower mantle pressures is also important to constrain the rheological behaviours of the lower mantle and subducting slabs. It has been predicted that the cold slab of peridotitic compositions, modelled by a mixture of $MgSiO_3$ bridgmanite and MgO, would become significantly soft due to the formation of inter-connection of the latter mineral with viscosity substantially lower than that of $MgSiO_3$ bridgmanite at relatively low temperatures in the slab[22,23]. However, the viscosities of these minerals under the lower mantle conditions have been estimated based on their homologous temperatures ($T/T_m$), which should have large uncertainties for MgO.

Here we determine the melting temperatures of MgO at pressures up to $\sim 50$ GPa using micro-texture observations of the quenched products in LHDAC by optical and scanning, and transmission electron microscopy, combined with the careful observations of the relation between the laser power and the temperature of the sample. We show two plateaus in the temperature increase with increasing input laser power, and interpret the first plateau is because of the significant plastic deformation of the sample together with Re gasket while the second one corresponds to the melting of the sample. The results of the earlier LHDAC study reporting a shallow melting curve for MgO, as well as those for some refractory metals[24,25], are likely attributed to miss-assignment of the former phenomenon to the melting of the sample, leading to a substantial underestimation of the melting temperatures under high pressure.

## Results

**Laser power versus temperature profiles.** We performed a series of LHDAC experiments using single crystal MgO as the starting material, which was loaded in a sample chamber together with an argon pressure medium. Temperature was determined by fitting each thermal radiation spectrum that was collected from the sample under laser-heating using the Planck's function (Fig. 1a and Supplementary Fig. 1)[26]. We first performed a melting experiment of MgO at ambient pressure to check the validity of the temperature measurement in our optic system. A clear temperature plateau was found at $3,010 \pm 140$ K in the power versus temperature profile (Fig. 1b), which agrees well with the reported melting temperature of MgO ($3,098 \pm 42$ K; ref. 27). Such a temperature plateau was also observed in the previous melting experiment using LHDAC[6] at high pressure and is thought to be usable for detecting melting of MgO at least at ambient pressure.

Figure 2 shows examples of the laser power versus temperature relationship obtained during heating in two separate runs at 32 and 33 GPa. In both cases, a plateau was observed in the range of 3,500–3,800 K of the power versus temperature profile (Fig. 2a). This temperature range corresponds to the melting temperature of MgO proposed by the earlier LHDAC study[6]. The radial temperature distribution across the MgO crystal under heating at this temperature range showed a standard Gaussian distribution (Fig. 2b). While the sample morphology did not clearly change during heating to the temperature just before the plateau (Fig. 2c,d), the sample quenched from the temperature plateau (point 'e' in Fig. 2a) was found to have expanded by $\sim 25$ % in the lateral direction (perpendicular to the compression axis) (Fig. 2e) along with the outward deformation of the sample chamber (gasket). The expansion of the sample chamber and sample was probably caused by the plastic flow of the Re gasket, which was induced at very high temperatures, as discussed later. Such a significant deformation (expansion) of the sample chamber was also observed in other runs performed at similar and lower/higher pressures (Supplementary Fig. 2), suggesting that this is an essential phenomenon in the present high temperature heating by $CO_2$ laser. When the sample was further heated to above 5,000 K, another plateau was observed in the power versus temperature profile (Fig. 2a). The temperature range of this second plateau is higher by 1,500–1,700 K than the first one and is well comparable to the theoretically predicted melting temperature of MgO (refs 13–15) at the corresponding pressure. The radial temperature distribution obtained during heating at the second plateau was found to become flat (Fig. 2b), compared with that obtained at lower temperatures. This flattening is likely due to a rapid increase in heat transfer caused by the convection of the MgO melt in a wide area, as discussed later. Unlike the case of the first plateau, neither deformation of the sample itself nor the sample chamber was observed.

**Micro-texture analyses.** Figure 3 shows typical examples of cross-sections of the samples recovered from two experiments in which they were heated to the temperatures of the first and second plateaus, respectively, and quenched (then decompressed). The laser-heated area (hot spot) of the sample quenched from the first plateau (Fig. 3a,b) consists of granular crystals of a few μm

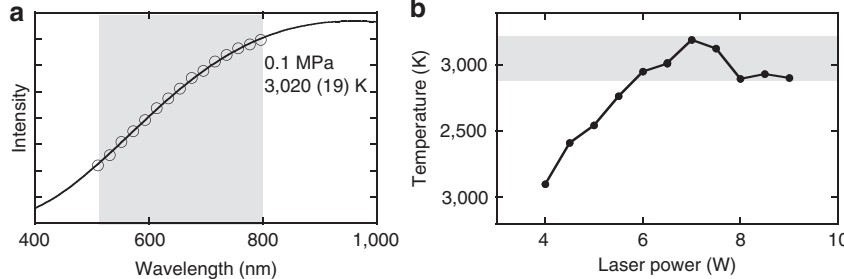

**Figure 1 | Typical thermal emission spectrum and temperature versus laser power profile of MgO at ambient pressure.** (**a**) Circles and a curve represent the radiation data and the fitting by the Planck's function, respectively. The grey shaded area shows the wavelength range in the fitting. The temperature estimated to be 3,020 K. (**b**) The grey shaded area shows the temperature range where a plateau was observed.

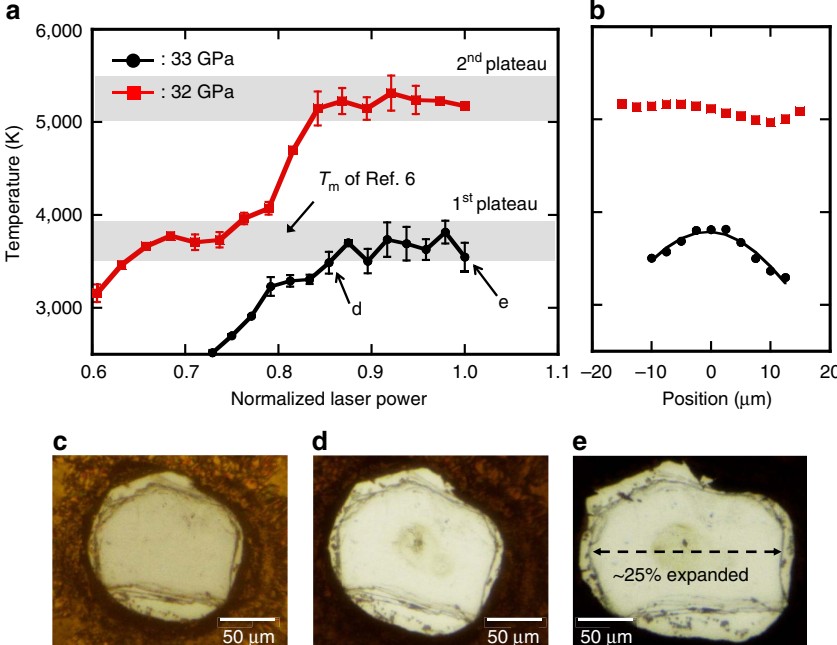

**Figure 2 | Temperature versus normalized laser power profiles together with radial temperature distributions and microscopic images.** (**a**) Circles and squares represent the profiles in laser heating at 33 and 32 GPa, respectively. The temperature errors were estimated by considering the difference between the temperatures obtained from the both sides of the cell. The grey shaded areas show the temperature ranges where plateaus were observed. (**b**) Circles and squares represent the temperature distributions in the MgO sample under heating at the first and second plateaus, respectively. The curve represents the fitting by the Gaussian function. (**c–e**) Microscopic images of the sample before (**c**) and after heating (**d,e**). The sample was found to have expanded by ∼25 % in the lateral direction after quenching from 3,700 K.

that are randomly aggregated as confirmed by electron diffraction in transmission electron microscope (TEM) (Fig. 3c). Many small voids were found at grain boundaries (mostly at triple junctions) and are likely remnants of liquid Ar that was formed and trapped during grain growth (recrystallization) of MgO at high temperature (Fig. 3c). On the other hand, in the low-temperature region (around the hot spot) the sample showed characteristic stripe patterns composed of significantly deformed and elongated domains as can be seen in the SEM image (Fig. 3d). TEM observation on a cross-section foil prepared from this region revealed that the individual domains are single crystal MgO with considerable amounts of dislocations, which are in contact with each other by low-angle grain boundaries (Fig. 3e). These features are evidence for the extensive shear deformation of the starting sample (single crystal MgO) during heating at the first plateau. The sample deformation was probably initiated with the thermally induced plastic flow of the gasket, which produced a large shear stress in the direction perpendicular to the compression axis. This also resulted in considerable thinning of the sample and

Ar pressure medium, significantly affecting the heating efficiency of the sample under laser irradiation. The observed first temperature plateau was most likely due to such flattening behaviour of the sample under deformation, which was also theoretically predicted to be caused by heterogeneous thermal stress in the sample[28]. Although such a significant deformation of sample/gasket was not observed in the earlier melting experiment on $(Mg,Fe)SiO_3$ perovskite, it was likely due to stress relaxation caused by the phase transition of the starting material (Supplementary Note 2).

On the other hand, the sample recovered from the second plateau (∼5,200 K at 32 GPa) showed a characteristic internal texture in the hot spot: a pair of layers composed of small granular crystals at the upper and lower edges and relatively large columnar crystals that are elongated perpendicular to the outer rims (Fig. 3f,g). This is well comparable to the solidification texture typically shown in metal casting[29]. The outermost layers are considered to be chilled margins in which multiple nucleation of equigranular crystals occurred on quenching of the MgO melt.

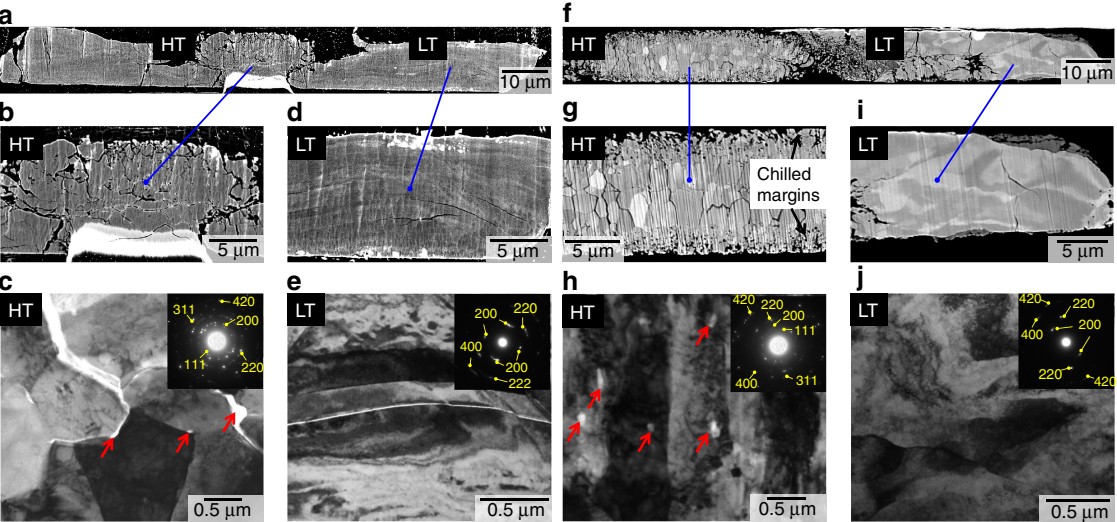

**Figure 3 | Cross-section images of the recovered samples.** (**a–e**) The sample recovered from 28 GPa and 3,800 K. (**f–j**) The sample recovered from 32 GPa and 5,200 K. (**a,b,d,f,g,i**) Orientation contrast images. (**c,e,h,j**) TEM bright field images. The region labelled 'HT' in each low-magnification image (**a,f**) is the laser-heated hot spot, while LT indicates the surrounding low-temperature region. The red arrows in **c,h** indicate small voids that are interpreted as remnants of liquid Ar (pressure medium) formed and trapped during grain growth of MgO at high temperature. Insets in **c,e,h,j** show electron diffraction patterns collected from each corresponding image, which are assigned to MgO (Supplementary Fig. 5, Supplementary Table 2 and Supplementary Note 1).

Subsequent to the formation of the chilled margins, larger columnar crystals grew inward from the rim due to the vertical heat flow in the sample chamber. It should also be noted that many small voids with spheroidal forms were found as inclusions within the individual columnar crystals (Fig. 3h and Supplementary Fig. 3). This fact also supports the direct solidification of the columnar crystals from MgO melt into which a part of Ar pressure medium was entrained as immiscible droplets. In the low-temperature region, the single crystal domains having a number of dislocations (Fig. 3i,j), similar to those observed in the low-temperature region of the sample quenched from the first plateau, were observed.

**Melting temperatures of MgO.** Figure 4 shows plots of our melting temperatures, $T_m$ of MgO together with the results of previous experimental studies and theoretical predictions. Fitting our $T_m$s to the Simon equation[30] yields $dT_m/dP$ of 103 K GPa$^{-1}$ at zero pressure, which is significantly higher than that obtained from the previous LHDAC experiments ($\sim$30 K GPa$^{-1}$; ref. 6). The $T_m$s of their study may have been substantially underestimated, since they are very close to the first temperature plateau in our study (Fig. 2), which was not caused by melting, but by plastic deformation of the sample (sample chamber). Note that the $dT_m/dP$ of our melting curve is consistent with those of the theoretical predictions (90–120 K GPa$^{-1}$; refs 13–15) as well as the estimation based on the Clausius–Clapeyron equation using the experimentally constrained volume and entropy changes through the melting ($\sim$100 K GPa$^{-1}$; ref. 31). Moreover, our melting curve agrees very well with that estimated based on Lindemann's relation using the experimentally determined Grüneisen parameter[32]. Extrapolation of the present melting curve of MgO to the pressure of the CMB (135 GPa) gives a melting temperature of $\sim$7,900 K, which is much higher than an expected geotherm temperature ($\sim$4,000 K) calculated from an outer-core adiabat and melting experiments on iron alloys[33].

## Discussion

The melting curve of MgO determined by the present experiments yields melting relations in the MgO–MgSiO$_3$ system

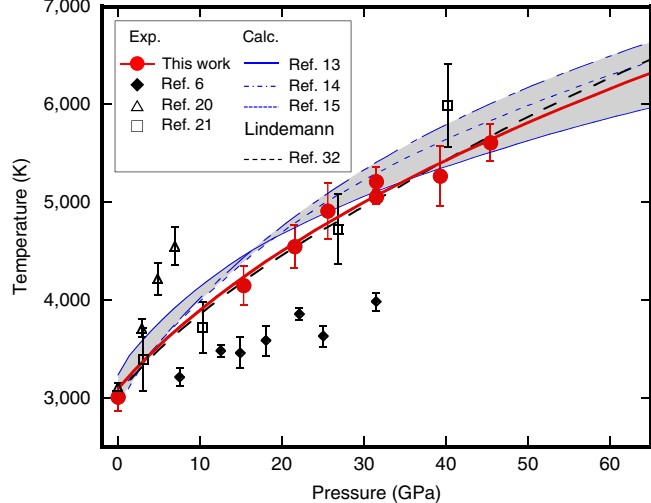

**Figure 4 | Melting curve of MgO.** The errors of our $T_m$s were estimated based on the s.d. of the temperatures at the second plateau. The details of our data are given in Supplementary Table 1. The grey shaded area represents the range of the melting curves predicted by the theoretical calculations[13–15].

at pressures throughout the lower mantle, which provides some insight into the chemical composition of the deep magma. For example, seismologically observed ultra-low velocity zone (ULVZ) near the core–mantle boundary (CMB) is believed to be caused by the partial melting of the lowermost mantle[1,7], where the melt composition can be inferred from the eutectic composition in the MgO–MgSiO$_3$ system. The previously reported gentle melting curve of MgO (ref. 6) results in an intersect with that of MgSiO$_3$ bridgmanite at $\sim$50 GPa[5], suggesting that the eutectic composition is located on the MgO-rich side relative to the peridotite composition at the CMB pressure[6,33]. However, the present steeper MgO melting slope offers eutectic compositions rather on the siliceous side throughout the lower mantle conditions. This means that the ultra-deep magmas responsible for the ULVZ are indeed

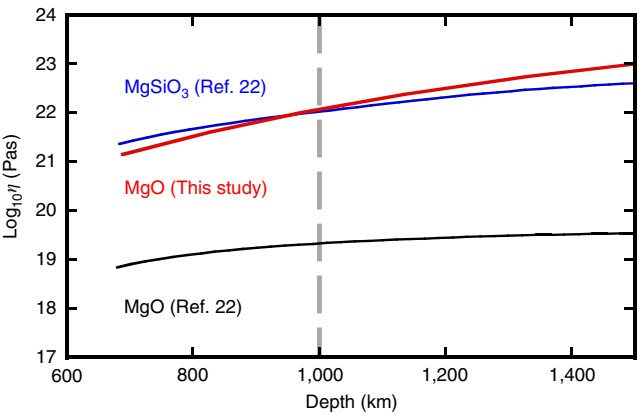

**Figure 5 | Viscosity-depth profile of the lower mantle.** Red, black, and blue curves were drawn based on the results of the present study (MgO) and a previous study[22] (MgO and MgSiO₃ bridgmanite). Our revised profile was calculated for the temperature gradient of 0.3 K km⁻¹ based on the homologous temperature scaling using our melting curve. The dashed line represents the depth where stagnations of subducting slabs are observed by seismic tomography[35,36].

peridotitic, as predicted by some earlier studies[18]. Taking into account the sink/float relationship between magma and solid mantle in the deep mantle[34], a considerable amount of iron (Fe/(Mg + Fe) > 0.24) is required to be incorporated into such peridotitic partial melt so that it is gravitationally stable to form the ULVZs at the bottom of the lower mantle. This scenario is also supported by the latest experimental study[9] on the Fe–Mg partitioning between the partial melt and solidus phases at the lowermost mantle conditions.

The rheological behaviour of the lower mantle is largely controlled by the viscosities of MgSiO₃ bridgmanite and MgO. The viscosity of MgO was estimated to be three orders of magnitude lower than that of MgSiO₃ bridgmanite based on the homologous temperature scaling using their melting curves[22]. This viscosity contrast between these minerals would cause rheological weakening of cold slabs subducted into the lower mantle because of the textural inter-connection of the softer MgO (ref. 23), resulting in slab stagnation at a depth of ~1,000 km as observed by seismic tomography[35,36] beneath some subduction zones[35,36]. However, the viscosity estimation by the homologous temperature scaling depends largely on the melting temperatures, $T_m$s, and the viscosity model[22] was calculated based on the significantly underestimated $T_m$s by the earlier LHDAC study[6] (Fig. 5). Recalculation using our refined $T_m$s yields much higher viscosity for MgO, almost equivalent to that of MgSiO₃ bridgmanite, under the lower mantle conditions (Fig. 5). This suggests that the rheological weakening would not occur in the cold subucting slabs, and the viscosity difference between the slab and the surrounding lower mantle may not be an important factor in the slab stagnation seismologically observed at ~1,000 km depth[35,36].

Determination of the melting temperature of refractory metals such as Mo, Ta and W at high pressure by LHDAC experiments is also difficult and remains controversial (for example, refs 24,25); the observed melting curves are significantly lower than those by shock-wave experiments[37] and theoretical calculations[38], as is the case for MgO. The result of the present study suggests a possibility that shear-induced anisotropic plastic flow[39] is also responsible for the anomalously low-temperature melting curves of these metals determined by the earlier LHDAC experiments. The present study demonstrates that the combination of *in situ* observation of the laser power versus

temperature relations and the detailed micro-texture analyses of the quenched sample is an effective and reliable approach for determining the melting temperatures of such refractory materials under very high pressure.

## Methods

**Starting material.** Single crystal MgO samples were loaded into a sample chamber with 100–150 μm diameter and 50 μm thick drilled in a preindented Re gasket in a diamond anvil cell (DAC). MgO sample was mechanically polished to adjust the thickness to 25–30 μm before loading. We used Ar pressure medium which provides chemically inert, thermally insulating, and virtually hydrostatic conditions. The sample was compressed to the target pressure at room temperature using flat top diamond anvils with 300 or 450 μm culets.

**High P–T experiment.** A $CO_2$ laser (10.6 μm, CW, TEM₀₀) system was adopted for the heating of the MgO sample which shows high absorption for the wavelength. Double-sided heating system equipped with a pair of 100 W $CO_2$ laser (Synrad, firestar t100) providing a Gaussian intensity distribution was used to reduce the temperature gradient across the axial direction of the sample chamber. The full width at half maximum (FWHM) of the heated (hot spot) region was approximately 60 μm, which was estimated by fitting the 1D temperature profile to the Gaussian function (Fig. 2b). The detail of the $CO_2$ laser-heating system is described in our previous report[26]. Temperature was determined by fitting the thermal radiation emitted from the sample under heating to the following Planck's function:

$$I(\lambda) = \frac{\varepsilon C_1 \lambda^{-5}}{e^{c_2/\lambda T} - 1},$$ (1)

where $I$ is measured intensity at each wavelength $\lambda$. $C_1$ and $C_2$ are constants and $\varepsilon$ is the emissivity, which is assumed to be independent of $\lambda$ in the observed wavelength range[33]. Each thermal radiation spectrum was measured from $2.5 \times 2.5$ μm² area at the centre of the heated region. For the Planck's fitting, the wavelength range from 500 to 800 nm was used. Figure 1a and Supplementary Fig. 1 show examples of the radiation spectrum obtained from MgO under heating at ambient pressure and at 18 GPa, respectively. Examples of the lateral temperature distribution are shown in Fig. 2b. The temperature at each laser power was determined by reading the peak value of the lateral temperature distribution profile. The temperature was determined by measuring the radiation from the both sides of the cell. The temperature uncertainty was given by considering the difference in temperature between the both sides. The melting temperature ($T_m$) at each pressure condition was determined by averaging the temperatures measured during the second temperature plateau (Supplementary Table 1). Pressures were estimated from the Raman shift of the diamond anvil culet[40] at room temperature after heating. For the run conducted at 9 GPa, the pressure was determined by the ruby fluorescence method[41].

**Micro-texture analyses.** The micro-texture analyses were made on quenched samples at Ehime University. After complete pressure release, the sample was recovered from the DAC and embedded in an epoxy resin. The cross-sections of the recovered sample were prepared by Ar ion milling using a cross section polisher (CP; JEOL, IB-19510CP). Field-emission scanning electron microscope (FE-SEM; JEOL, JSM-7000F) was used for micro-structure analysis. The size and shape of individual MgO crystals were examined by electron channelling contrast imaging at an accelerating voltage of 5 kV and a beam current of 3 nA using a back-scattered electron detector. After SEM observation, TEM cross-section foils of ca. $10 \times 5$ μm × 0.1–0.2 μm thick were cut out from the target areas using a focused ion beam (FIB; JEOL, JEM-9310FIB) system. TEM (transmission electron microscope) and analytical scanning-TEM (STEM) observations were performed at an accelerating voltage of 200 kV using a field-emission TEM (JEOL, JEM-2100F) equipped with high sensitive CCD cameras (Gatan, Orius 200D and UltraScan 1000XP) and a silicon drift detector (JEOL, JED-2300). Scanning-TEM (STEM)-EDS elemental mapping analysis was performed on a $8 \times 8$ μm² area of some cross-section foils using a spot size of 0.5 nm.

**Estimation of thermal pressure.** Since the temperature range of our interest is so high that we cannot rule out the thermal pressure effect, we adopted the Mie–Grüneisen equation[42] to estimate the contribution of thermal pressure ($P_{th}$).

$$P_{th} = \frac{\gamma}{V}(E_{th} - E_{th}300),$$ (2)

where $\gamma$ is the Grüneisen parameter, $V$ is the volume, $E_{th}$ is the thermal energy and $E_{th}300$ is the thermal energy at 300 K. On the basis of the Debye model, the $E_{th}$ at a given temperature can be expressed as

$$E_{th} = 9nRT\left(\frac{\Theta_D}{T}\right)^{-3} \int_0^{\Theta_D} \frac{x^3}{e^x - 1} dx,$$ (3)

where $R$ is the gas constant, $n$ is the number of atoms in the formula unit of the concerned material ($n = 2$ for MgO), $T$ is the temperature and $\Theta_D$ is the Debye

temperature. Since the temperature range used in this study is much higher than the Debye temperature of MgO ($\Theta_D = 936$ K), we can take the high temperature Dulong and Petit limit for $E_{th}$ by following[43]. The $P_{th}$ is finally expressed as

$$P_{th} = \frac{6\gamma RT}{V} - \frac{\gamma}{V}(E_{th}300). \tag{4}$$

We used the Grüneisen parameter constrained by the equation of state (EOS) data obtained by both static and dynamic experiments[32]. Substituting the thermal EOS data, which were obtained by laser-heated diamond anvil cell experiments using a $CO_2$ laser by Fiquet et al.[44], into equation (4), we obtained the $P_{th}$ as a function of the temperature (Supplementary Fig. 4). By linearly fitting the obtained $P_{th}$s, the $P_{th}$ can be approximately expressed as

$$P_{th} = -0.27 + 1.66 \times 10^{-3}T. \tag{5}$$

The pressure range studied by Fiquet et al.[44] is from 3.2 to 15.7 GPa, which is comparable to that of our study (8.7–36.3 GPa). We used the above equation (5) to estimate the $P_{th}$s of our $T_m$s, and the results are shown in Supplementary Table 1.

**Melting curve of MgO.** The MgO melting curve was determined by fitting the $T_m$s obtained in this study to the Simon equation:

$$T_m = T_0 \left( \frac{P_m}{A} + 1 \right)^{\frac{1}{C}}, \tag{6}$$

where $T_0$ is the melting temperature at zero pressure, $P_m$ is the pressure at melting, and the coefficients $A$ and $C$ are constants[30]. We obtained $A = 11.1$ (4.0) and $C = 2.7$ (0.5) by using a reported $T_0$ value of $3,098 \pm 42$ K (ref. 27). The melting extrapolated to 135 GPa is 7,900 K with an uncertainty of the order of a thousand K. The estimated melting curve is in very good agreement with that estimated based on the Linderman's relation using an experimentally constrained Grüneisen parameter[32] (Fig. 4), which was also used for the thermal pressure estimation in this study.

**Data availability.** No data sets were generated or analysed during the current study, but the original data in this article are available from the corresponding author on reasonable request.

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

## Acknowledgements

We thank Y. Kuwayama and S. Ohnishi for their assistance in the high-pressure experiments. Discussions with T. Sakai, T. Ohuchi, T. Yagi, Z. Du and T. Taniushi were helpful. This work is funded by the JSPS Grant-in-Aid for Scientific Research

(Nos JP25220712, JP24740356, JP16K05612, and JP16J00612). This work was supported by the Joint Usage/Research Center PRIUS, Ehime University. This work was partly supported by the joint research project (No 2014B2-34) of the Institute of Laser Engineering, Osaka University.

## Author contributions

T.K. organized the research project, conducted experiments, analysed the data, and completed the manuscript with the help of H.O., M.N.; and T.I. T.K. and H.O. carried out the micro-texture analyses. All authors discussed the results and developed the manuscript.

## Additional information

**Competing interests:** The authors declare no competing financial interests.

