## [Peer Review File · Nature Communications]

Reviewer #1 (Remarks to the Author):

This manuscript by Kimura and others reported the melting point of the MgO, which is the second abundant mineral in the lower mantle, under high pressure condition by using experimental method.

The authors address important issue, but I feel that the current manuscript is too specific to be published in Nature Communications. I can recommend this paper for the publication only if the manuscript is substantially re-written with additional experimental data.

Weak implication

Based on the experimental data, the authors discussed about ULVZ, which is observed at core-mantle boundary region by seismology. Unfortunately, current manuscript does not provide any new insight about ULVZ. The melting temperature of MgO is higher than the previous LHDAC experiments and is consistent with theory. So what? How the expected nature of ULVZ or deep magma will be changed by the new data? Current manuscript made no effort even to extrapolate the melting temperature to CMB condition.

Experimental data

Experimental pressure ($\sim < 30$ GPa) is far away from CMB condition (135 GPa). I would expect experimental pressure will be extended in the revised manuscript. Otherwise, numerous data are required to reduce the uncertainty of extrapolation.

Did authors obtain 1-D or 2-D mapping of the temperature distribution (e.g., Du and Lee 2014 GRL)?

The authors argue that the melting temperature may have large uncertainty when it was determined by the criteria using plateau in W-T curve. I agree with this point, but I can not be convinced by the scenario provided in the discussion part (line 64-81). I can not understand how the deformation and the plateau are correlated each other. Moreover, the deformation of the sample chamber is often observed at around < 30 GPa. This is another reason why high-P data are necessary.

Reviewer #2 (Remarks to the Author):

The paper presents an experimental study of the melting curve of MgO up to 32 GPa, based on diamond anvil cell (DAC) and laser heating. The main result of the paper is a new estimate of the melting curve, which is much higher than that reported in a previous DAC study (Ref. 6), and agrees well with shock data and results obtained from theoretical calculations.

The authors also offer an explanation as to why the early results reported in Ref. 6 may have underestimated the MgO melting curve, which would be down to the incorrect identification of the onset of melting. The criterium used to locate the melting transition is the plateau in the temperature vs power curve: when the system melts the temperature stops increasing for a while, due to the latent heat of melting.

However, the authors observe two plateaus in their data. When they recover the sample after reaching the first plateau, located at the lower temperature, they find that the sample has expanded in the direction perpendicular to the axial compression, showing plastic flows. A TEM analysis of the recovered sample also showed stripe patterns, characteristic of shear deformation. The resulting thinning of the sample could have affected the efficiency of the heating, causing the plateau. On the other hand, the sample recovered after reaching the second plateau, at the higher temperature, showed the characteristics of a recrystallized sample, with Ar droplet inclusions which have

immiscibly dissolved into the liquid sample.

he data are convincing, and appear to resolve an outstanding issue with the melting behaviour of MgO. The authors also point out that there is a whole class of transition metals for which DAC experiments also appear to measure relatively low melting curves, or more specifically low melting slopes, and

they suggest that also these experiments may have been affected by wrong identification of the onset of melting. In fact, similar suggestions were recently proposed (Anzellini et al, 2013) for the melting curve of Iron, where large discrepancies existed between some early DAC experiments on one side (Boehler) and shock data (Holmes) and ab-initio calculations (Alfe`) on the other. The paper by Anzellini et al. also based on DAC, reconciled these discrepancies. Iron of course is very important for the deep Earth, being the main component of the Earth's core. It might be more appropriate therefore to mention the iron story rather than the molybdenum one, or at least add a mention to the iron story.

The other comment that I would make is that it is somewhat far fetched to extrapolate the present results, up to 32 GPa, to the core mantle boundary pressure of 135 GPa, and so the authors may want to add some words of caution.

I would recommend publication of the work once the authors have addressed the above comments.

Reviewer #3 (Remarks to the Author):

Review of Article "Melting temperatures of MgO under high pressure by micro- texture analysis" by T. Kimura, H. Ohfuji, M. Nishi, and T. Irifune [Manuscript # NCOMMS-16-11983]

This paper investigates the melting behavior of MgO periclase to pressures of the upper portions of the lower mantle using the laser-heated diamond-anvil cell (LHDAC). The high-pressure melting curve of MgO has been contentious over the past couple of decades yielding disagreement within and between experimental and theoretical studies. When extrapolating to the core-mantle boundary (CMB), the differences between studies can be several thousands of degrees different! This is an important problem, but I'm not entirely convinced of the data even though the logic makes sense and the data appear fairly consistent with recent melting estimates of MgO from melting of ferropiclase (Mg,Fe)O.

The authors are using two methods to infer melting of MgO: laser power-temperature relationship and quenched micro-texture. While these are common ways that some studies have inferred melt, the methods have problems, as they show in their manuscript. Quenched texture may reflect solid-solid structural phase transitions or even "shear-induced plastic flow." Discontinuities in laser power vs. temperature profiles, used frequently to infer melting temperature, may also be caused by recrystallization or grain growth prior to melting. They did not mention, perhaps on purpose, that many think that the plateau in laser power vs. temperature may be due to latent heat, although this has been shown to not likely be the case (e.g., Geballe & Jeanloz, 2012).

Their experiments are similar to those run by (Zerr & Boehler, 1994) in which they use single-crystal MgO in an Ar pressure medium/thermal insulation and heating by CO2 laser. In the earlier study, Zerr & Boehler used the visual observation of a large increase in temperature to mark the onset of melting inferring that the melt could now absorb the laser energy more readily. The current study uses temperature plateaus and quenched texture. The initial test of the system of melting MgO at room pressure yielded a promising temperature in that it was very similar to literature values. More information is needed on this test. Was the melting done in air or under an inert atmosphere? Is there chemical information of the quenched sample (does the sample oxidize

or hydrate)? What does the texture look like?

Now moving on to the experiments at high pressures. In these experiments, the authors claim to observe two plateaus in the laser-power vs. temperature profiles: the first they infer to be when the Re gasket flows due to weakening of the material under high temperatures. A second plateau was observed some 1500-1700 degrees greater than the first plateau. This plateau, they claim, is when the sample melts. But in lines 79-81, they say "no clear change was observed" in the sample chamber or sample morphology. But I thought the premise of this study was to look at the micro-texture upon quench?

This takes me to Figure 3 in which back-scattered electron (BSE) images and TEM images are shown. BSE gives info on atomic number Z: typically brighter regions are represented by higher Z material thus giving a sense of composition. As such I'm confused on what I'm looking at in the BSE images. Why the color change in Figure 3i or between the crystals in HT? Shouldn't the color be the same throughout given it is only MgO? Or is there contamination from Ar? Water? Carbon (diamond)? Or something else? In Figure 3a,b,d, we see very different textures in HT and LT for what should be all be below the melting temperature. In Figure 3f,g,i, we see much different texture. Additionally, what is happening in the middle of Figure 3f, the region between "HT" and "LT"? The "chilled margins" make sense since that is likely self-insulation layers from thinning Ar layers, but there isn't such a feature in the other regions? Why not?

Why are the "HT" regions so different in extent? It looks like a factor of 2. Is this reasonable? What are the expected temperature gradients between the "HT" and "LT" regions? How long did the heating experiments last? Did they try to just heat to at high laser power, rather than ramping up to see if the texture was the same with melting, rather than an effect of just grain growth with time?

What do the electron diffraction images show? Is it MgO? Or something else?

Now on to thermal pressure... This is tricky. There have been several studies that have claimed that thermal pressures are negligible when using a soft pressure medium such as Ar (e.g., Fischer et al, 2013; Zerr & Boehler, 1994) and others that suggest that $P_{th} \sim 0.5 \alpha K \Delta T$ (e.g., Goncharov et al., 2010), thus when ΔT is large, P_{th} may also be large. In any case, when comparing to Zerr & Boehler, who didn't add P_{th} , the agreement becomes even worse since Zerr & Boehler do not include thermal P.

Almost as an aside, the authors make mention of the controversies in melting temperature in refractory metal by LHDAC experiments. The possibility that it is occurring due to "shear-induced anisotropic plastic flow" is an intriguing idea... But shouldn't this happen along the same P/T path for all samples since it would be dictated by the Re gasket? If this method proves to be reliable (I'm not yet convinced), this may be good to state it in the discussion, but not the abstract as it is off topic.

Figure 2b, c, d: How much time was the sample heated for between c and d? The temperature holds steady, but as the gasket got weak, the sample appear to expand. What is causing the browning of the sample? Did the diamonds burn (especially shown in part d)?

Figure 3: The agreement with Du et al. (2014) is rather striking, although the estimates of the melting temperature at the CMB are different by nearly 1000 K (8000 vs 8900 K). Why the discrepancy?

Figure S3: There appears to be a slight shift in the MgO peaks between the "HT" and "LT" regions. What do you attribute this to? Are the MgO (and Ar) volumes consistent for this pressure? What are the uncertainties? Why are the "HT" peaks broader?

Supplementary Table S2: I'm very puzzled at the relevance on the values given in this table. The average over such a large region (20 μm x 20 μm) is problematic. I'd prefer to see a compositional map. Is the melt region enhanced in Ar or the other way around? How was Ar quantified? What were the "real" totals before normalization?

Minor problems:

Line 9: add "most" after, "Periclase (MgO) is the second..."

Line 10: remove "chemical". This is redundant.

Line 11: "mantle-core boundary" should be "core-mantle boundary" to go with standard convention.

Line 175: missing "nm" after "500 to 800"

I really want to like this paper, but unfortunately I am not convinced by the data shown.

**Response to review of the manuscript “Melting temperatures of MgO**
**under high pressure by micro-texture analysis”**

The reviewer’s comments are shown in blue and our responses in black color.

**Response to Referee #1**

>This manuscript by Kimura and others reported the melting point of the MgO, which is the
second abundant mineral in the lower mantle, under high pressure condition by using
experimental method. The authors address important issue, but I feel that the current manuscript
is too specific to be published in Nature Communications. I can recommend this paper for the
publication only if the manuscript is substantially re-written with additional experimental data.

We made efforts to expand our temperature and pressure range toward ~6000K and
~50GPa, respectively, which are certainly the limits in the current state of the art of the LHDAC
technology combined with CO₂ laser heating, and extensively re-written the manuscript
according to the referee’s recommendation (see below).

>Based on the experimental data, the authors discussed about ULVZ, which is observed at
core-mantle boundary region by seismology. Unfortunately, current manuscript does not
provide any new insight about ULVZ. The melting temperature of MgO is higher than the
previous LHDAC experiments and is consistent with theory. So what? How the expected nature
of ULVZ or deep magma will be changed by the new data? Current manuscript made no effort
even to extrapolate the melting temperate to CMB condition.

As experimental scientists, we believe in experimental data rather than theoretical
predictions; as for the melting temperatures of MgO, at high pressure, there have been only one
report by Zerr and Boehler (1994) up to 30 GPa, for which we found a serious problem in their
criteria for determination of melting temperatures. Determination of the melting temperatures
by careful examination of micro-textures of the run products combined with the conventional
method of using the temperature-laser power relations yielded substantially higher melting
temperatures of MgO by about 1000K at ~30 GPa than those of the earlier study, which is really
new and has significant implications for the determination of melting temperatures of other
refractory materials, as discussed in the text. Moreover, we have just succeeded to expand the

pressure region to ~50 GPa, which further confirm our results in the earlier version of our paper,
demonstrating that our experimental results should be used for the melting curve of MgO. It is
fortunate that our data are quite consistent with the theoretical predictions and dynamic
compression data, but we believe carefully determined experimental data are most valuable in
scientific research.

As for the implications of our new data, we added an important point in Discussion on the
rheological properties of lower mantle materials, which have been constrained based on the
homologous temperatures (T/T_m) of the constituting major minerals of MgSiO₃ bridgmanite and
MgO (line 168-182). We have extrapolated our melting temperatures to the bottom of the lower
mantle, and discussed the expected melting relations and the composition of the ultra-deep
magma to be formed near the mantle-core boundary, which are quite different from those based
on the earlier melting experiments by Zerr and Boehler (1994) (see, line 151-167).

Experimental data

>Experimental pressure (~< 30 GPa) is far away from CMB condition (135 GPa). I would
expect experimental pressure will be extended in the revised manuscript. Otherwise, numerous
data are required to reduce the uncertainty of extrapolation.

It is extremely challenging to determine the melting temperatures of such highly refractory
materials as MgO under high pressure, which has a melting temperature as high as 3098 K even
at the ambient pressure (Ref. 27). In addition to our new technique of the both-sided CO₂ laser
heating system, which allows us to generate such high temperatures at high pressure, as shown
in Supplementary Fig. 3 (Ref. 26), we adopted the triplet lens specially designed for correction
of chromatic aberration (Edmund, 64838-L) as the objective lens. This technique allows us to
accurately measure the temperature as shown in Fig. 1a and Supplementary Fig. 1, and we have
eventually succeeded in determining the T_m up to ~50 GPa, which is substantially higher than
those of earlier LHDAC experiments and more tightly constrains the melting curve of MgO as
shown in Fig. 4 of the revised manuscript. We should point out that our new data that added in
this figure are very consistent with our results for the experiments to ~30GPa, confirming our
earlier conclusion on the steep gradient of the melting curve of MgO.

>Did authors obtain 1-D or 2-D mapping of the temperature distribution (e.g., Du and Lee 2014
GRL)?

We added a typical example of the 1-D temperature profile as Supplementary Fig. 2 to the
revised manuscript. The full width at half maximum (FWHM) of the heated region was
approximately 60 μm , which was estimated by fitting to the Gaussian function, as described in
line 205-207. The temperature difference in the center of the heated area of $2.5 \times 2.5 \mu\text{m}^2$ was
~ 100 K, obtained from the results of the 1-D profile, which is significantly less than the
averaged-measurement error of ~ 200 K.

**Supplementary Figure 2. Typical example of the radial temperature distribution in**
**the MgO sample under heating at 36 GPa.** The plots and curve represent the
measured temperature and the fitting by the Gaussian function.

>The authors argue that the melting temperature may have large uncertainty when it was
determined by the criteria using plateau in W-T curve. I agree with this point, but I can not be
convinced by the scenario provided in the discussion part (line 64-81). I can not understand how
the deformation and the plateau are correlated each other. Moreover, the deformation of the
sample chamber is often observed at around <30 GPa. This is another reason why high-P data
are necessary.

As for the question about the relation between the sample/gasket deformation and the
temperature plateau, the expansion of the sample in the lateral direction (Fig. 2d), which was
induced by the plastic flow of the gasket and gradually proceeded during heating at the first

temperature plateau (3500-3800 K), resulted in a considerable thinning of the sample and Ar
pressure medium. The reduction in thickness of the sample chamber (along the compression
axis of the DAC) must have had a significantly negative effect on the heating efficiency, as
described in line 113-120.

We are not sure what the reviewer means by “the deformation of the sample chamber is
often observed at around <30 GPa”, but what we are sure is that such an extensive deformation
of the sample chamber is not observed commonly in LHDAC experiments, where the sample is
heated to “moderate” temperatures (2000-2500 K), and so far, has not been reported anywhere
as far as we know. In the additional melting experiments at higher pressures which we newly
conducted, we also observed the deformation of the sample together with the Re gasket at the
first plateau, as shown in Supplementary Fig. 3, suggesting that it is an essential phenomenon in
the present high temperature heating.

**Response to Referee #2**

>The paper presents an experimental study of the melting curve of MgO up to 32 GPa, based on
diamond anvil cell (DAC) and laser heating. The main result of the paper is a new estimate of
the melting curve, which is much higher than that reported in a previous DAC study (Ref. 6),
and agrees well with shock data and results obtained from theoretical calculations.

The authors also offer an explanation as to why the early results reported in Ref. 6 may have
underestimated the MgO melting curve, which would be down to the incorrect identification of
the onset of melting. The criterion used to locate the melting transition is the plateau in the
temperature vs power curve: when the system melts the temperature stops increasing for a while,
due to the latent heat of melting. However, the authors observe two plateaus in their data. When
they recover the sample after reaching the first plateau, located at the lower temperature, they
find that the sample has expanded in the direction perpendicular to the axial compression,
showing plastic flows. A TEM analysis of the recovered sample also showed stripe patterns,
characteristic of shear deformation. The resulting thinning of the sample could have affected
the efficiency of the heating, causing the plateau. On the other hand, the sample recovered after
reaching the second plateau, at the higher temperature, showed the characteristics of a
recrystallized sample, with Ar droplet inclusions which have immiscibly dissolved into the
liquid sample. The data are convincing, and appear to resolve an outstanding issue with the
melting behaviour of MgO.

We thank the referee, who adequately addresses the important points in our paper.

The authors also point out that there is a whole class of transition metals for which DAC
experiments also appear to measure relatively low melting curves, or more specifically low
melting slopes, and they suggest that also these experiments may have been affected by wrong
identification of the onset of melting. In fact, similar suggestions were recently proposed
(Anzellini et al, 2013) for the melting curve of Iron, where large discrepancies existed between
some early DAC experiments on one side (Boehler) and shock data (Holmes) and ab-initio
calculations (Alfe') on the other. The paper by Anzellini et al. also based on DAC, reconciled
these discrepancies. Iron of course is very important for the deep Earth, being the main
component of the Earth's core. It might be more appropriate therefore to mention the iron story
rather than the molybdenum one, or at least add a mention to the iron story.

We agree that the earlier work by Boehler on the melting temperatures of Fe could have
been also underestimated because of a similar reason as discussed here. However, we note the
first plateau occurs at temperatures between 3500-4000K, in the pressure range studied here.
The melting temperatures of Fe determined by this author is significantly lower than these
temperatures, and may not have directly relevance to the present phenomenon of plastic flow of
Re gasket. Actually, a recent study using X-ray absorption spectroscopy coupled with DAC
proposed lower melting temperatures, which is consistent with those of Boehler (G. Aquilanti et
al., Proc. Natl. Acad. Sci. 2015). This is the reason why we limit our discussion on the
refractory metals whose melting temperatures higher than ~3000K at the ambient condition.

The other comment that I would make is that it is somewhat far fetched to extrapolate the
present results, up to 32 GPa, to the core mantle boundary pressure of 135 GPa, and so the
authors may want to add some words of caution.

In response to the reviewer's comment, we have eventually determined the T_m up to ~50
151 GPa, which is the highest pressure in the LHDAC experiments relevant to melting temperatures
of MgO, as shown in the revised Fig. 4. The results well agree with the extrapolation based on
our earlier runs up to ~30 GPa, and we are confident our melting temperature can be
extrapolated to further higher pressures. Nevertheless, it is true that there is uncertainties of

~1000 K in our estimated melting temperature (7900K) at the mantle-core boundary pressure,
which is clearly stated in line 269-270.

I would recommend publication of the work once the authors have addressed the above
comments.

We thank the referee for the great efforts to carefully check all the materials involved.

**Response to Referee #3**

>Review of Article "Melting temperatures of MgO under high pressure by micro- texture
analysis" by T. Kimura, H. Ohfuji, M. Nishi, and T. Irifune [Manuscript #
NCOMMS-16-11983]

>This paper investigates the melting behavior of MgO periclase to pressures of the upper
portions of the lower mantle using the laser-heated diamond-anvil cell (LHDAC). The
high-pressure melting curve of MgO has been contentious over the past couple of decades
yielding disagreement within and between experimental and theoretical studies. When
extrapolating to the core-mantle boundary (CMB), the differences between studies can be
several thousands of degrees different! This is an important problem, but I'm not entirely
convinced of the data even though the logic makes sense and the data appear fairly consistent
with recent melting estimates of MgO from melting of ferropericlase (Mg,Fe)O.

>The authors are using two methods to infer melting of MgO: laser power-temperature
relationship and quenched micro-texture. While these are common ways that some studies have
inferred melt, the methods have problems, as they show in their manuscript. Quenched texture
may reflect solid-solid structural phase transitions or even "shear-induced plastic flow."
Discontinuities in laser power vs. temperature profiles, used frequently to infer melting
temperature, may also be caused by recrystallization or grain growth prior to melting. They did
not mention, perhaps on purpose, that many think that the plateau in laser power vs. temperature
may be due to latent heat, although this has been shown to not likely be the case
(e.g., Geballe & Jeanloz, 2012).

First, we should point out that the micro-texture analysis of the recovered sample,
especially the one including the cross section, is not a common but the state of the art technique

for melting experiments using LHDAC, in spite of the referee's comment. In fact, this method
has been adopted in only very few recent works (Refs. 9 and 21) for the determination of
melting relations of multi-component systems, where the melting temperatures of the samples
were estimated from a combination of micro-texture and chemical composition analyses.

The point in this study is that for accurate determination of the melting temperature of
refractory, single-component material under high pressures, careful evaluation of melting
criteria is essential and important, such as by combining the observation of plateau(s) in
temperature-power relation with the micro-texture analysis of the quenched product. In the
present study, we found two plateaus, the first plateau attributed to the sample deformation
(flattening) initiated by the plastic flow of the Re gasket, and second plateau caused by melting
of the sample. These are also supported by microtexture features shown by each quenched
product: the former shows a mosaic texture formed by significantly deformed (elongated)
crystals involving many dislocations and subgrain boundaries (Fig. 3a, d, and e), while the latter
shows a typical quenching texture composed of chilled margins and columnar crystals grown
inward from the rim (Fig. 3f and g). The presence of small Ar inclusions within the MgO
crystals quenched from the second plateau (Fig. 3h and Supplementary Fig. 4) is also evidence
for melting, where a part of Ar pressure medium was entrained as immiscible droplets in MgO
rapidly crystallized from the melt, as described in the text (line 128-132). A drastic change of the
heat transfer rate due to the convection of the MgO melt may be the cause for the second
plateau, but further study is needed for detail.

>Their experiments are similar to those run by (Zerr & Boehler, 1994) in which they use
single-crystal MgO in an Ar pressure medium/thermal insulation and heating by CO2 laser. In
the earlier study, Zerr & Boehler used the visual observation of a large increase in temperature
to mark the onset of melting inferring that the melt could now absorb the laser energy more
readily. The current study uses temperature plateaus and quenched texture. The initial test of the
system of melting MgO at room pressure yielded a promising temperature in that it was very
similar to literature values. More information is needed on this test. Was the melting done in air
or under an inert atmosphere? Is there chemical information of the quenched sample (does the
sample oxidize or hydrate)? What does the texture look like?

The melting test of MgO at ambient pressure was performed in the air using the same
optical setting. SEM observation of the laser-heated spot of the sample quenched from ~3100 K,

where we clearly observed a plateau in the laser-power vs temperature profile, showed a crater
and many cubic MgO crystals grown in the surroundings. Such euhedral MgO crystals most
likely precipitated from the melt upon quenching. According to the reviewer's request, we
performed quantitative analysis of the laser-heated and also surrounding areas by SEM-EDS
and confirmed that they consist exclusively of pure MgO (Mg 38.85 wt%, O 60.59 wt%, Total
99.44 wt%) and neither secondary phase(s) nor impurities were observed. We also checked with
micro-Raman spectroscopy and detected no signs of secondary phase(s) such as brucite
$\text{Mg}(\text{OH})_2$ detected in the areas, as described in our revised Supplementary Notes in line 66-69.

**Figure I. Back-scattered electron image of the MgO sample quenched from ~3100 K at**
**ambient pressure.**

>Now moving on to the experiments at high pressures. In these experiments, the authors claim
to observe two plateaus in the laser-power vs. temperature profiles: the first they infer to be
when the Re gasket flows due to weakening of the material under high temperatures. A second
plateau was observed some 1500-1700 degrees greater than the first plateau. This plateau, they
claim, is when the sample melts. But in lines 79-81, they say "no clear change was observed" in
the sample chamber or sample morphology. But I thought the premise of this study was to look
at the micro-texture upon quench?

There are indeed clear differences in microtexture between the samples quenched from the
first and second plateaus, as shown in Fig. 3 (line 101-134). However, they can only be seen in
the cross-sections of the recovered samples but are not clearly recognized for the sample after
laser-heating in the DAC. This is a reason why our direct observation of the quenched samples
on their cross-sections is important to judge melting in addition to the (second) temperature

plateau. However, to avoid any confusion, we revised the sentence to “Unlike the case of the
first plateau, neither deformation of the sample itself nor the sample chamber was observed.”
(line 99-100).

>This takes me to Figure 3 in which back-scattered electron (BSE) images and TEM images are
shown. BSE gives info on atomic number Z: typically brighter regions are represented by higher
Z material thus giving a sense of composition. As such I'm confused on what I'm looking at in
the BSE images. Why the color change in Figure 3i or between the crystals in HT? Shouldn't the
color be the same throughout given it is only MgO? Or is there contamination from Ar? Water?
Carbon (diamond)? Or something else?

There seems to be a confusion here. The SEM images in Figure 3 show electron
channeling contrast which is produced as a result of the interaction of energetic electrons with
crystalline materials and is very sensitive to its crystallographic orientations (Newbury et al.,
1986), as we mention in the method section. Thus, the variation in contrast in the images
reflects the variety of crystallographic orientations of the MgO crystals in the product, which is
helpful for us to recognize the individual grain size and shape. Images taken by using
back-scattered electron (BSE) detector also have information on atomic number Z as the
reviewer pointed out and are indeed widely used to discuss the chemical features of materials.
In the present study, we used a relatively low accelerating voltage (5 kV) and high beam current
(3 nA) to enhance the contribution of crystallographic orientation contrast. We also checked by
SEM-EDS mapping analysis that there is neither chemical contamination (such as by Ar or C)
nor impurities in the samples, although small Ar inclusions were found to be present within
individual crystals and their boundaries by STEM-EDS.

To avoid such confusion, we modified the caption of Fig. 3 by replacing the phrase
“Back-scattered electron images” with “Orientation contrast images”.

>In Figure 3a,b,d, we see very different textures in HT and LT for what should be all be below
the melting temperature. In Figure 3f,g,i, we see much different texture. Additionally, what is
happening in the middle of Figure 3f, the region between "HT" and "LT"? The "chilled
margins" make sense since that is likely self-insulation layers from thinning Ar layers, but there
isn't such a feature in the other regions? Why not?

>Why are the "HT" regions so different in extent? It looks like a factor of 2. Is this reasonable?
What are the expected temperature gradients between the "HT" and "LT" regions? How long
did the heating experiments last? Did they try to just heat to at high laser power, rather than
ramping up to see if the texture was the same with melting, rather than an effect of just grain
growth with time?

According to the temperature distribution measured across the hot spot (Supplementary Fig.
2), the temperature in the LT region is estimated to be ~3000 K when the temperature in the HT
region reached at ~5000 K. Since laser-heating at such high temperatures (which is needed for
the melting of MgO) may lead to failure of the diamond anvil(s), the heating duration at each
laser power had to be short (only for several seconds), then the total heating duration over the
first and second plateaus is ~1 min each. However, it is clear from the microtexture that such a
short-time heating is adequate to judge melting.

In the both samples quenched from the first and second plateaus, the microtexture of
MgO is indeed quite different between the HT and LT regions. This is simply due to the large
temperature gradient horizontally across the whole sample chamber, since the hot spot is much
smaller than that of the sample hole (~ 120 μm), as shown in the 1-D temperature profile
(Supplementary Fig. 2). The LT regions of the both samples (quenched from the first and
second plateaus) consist of significantly deformed crystals due to the large shear stress induced
by the gasket flow during heating, in which the temperature was, however, not high enough to
promote the recrystallization and grain growth. More details are described in line 108-116.

On the other hand, the microtexture of the HT regions is quite different between the two
samples. The sample quenched from the first plateau shows recrystallization and grain growth
features of MgO, while that from the second plateau shows a typical quenching texture from the
MgO melt. The latter is characterized by chilled margins and elongated crystals grown inward
from the chilled margins (Fig. 3f and g). Originally, the chilled margins were continuous along
the periphery of the lens-shaped melted region, but are now seen only at the lower and upper
edges (Fig. 3f and g). The porous area in the middle of Fig. 3f is also a chilled margin composed
of extremely fine grains of MgO, but it has not been well-polished due to drop-out of many
grains, since this region was not sufficiently supported by the epoxy resin compared with the
rim parts (that is directly in contact with the resin).

>What do the electron diffraction images show? Is it MgO? Or something else?

The electron diffraction patterns (shown as insets of Figs. 3c, e, h and j) were obtained
from the center of the corresponding bright-field images using a selected-area aperture of a 1.4
315 μm diameter. All the observed diffraction spots (and discontinuous rings) are assigned to MgO,
as indexed in the revised Fig. 3.

>Now on to thermal pressure... This is tricky. There have been several studies that have claimed
that thermal pressures are negligible when using a soft pressure medium such as Ar (e.g.,
Fischer et al, 2013; Zerr & Boehler, 1994) and others that suggest that $P_{\text{th}} \sim 0.5 \alpha K \Delta T$
(e.g., Goncharov et al., 2010), thus when ΔT is large, P_{th} may also be large. In any case, when
comparing to Zerr & Boehler, who didn't add P_{th} , the agreement becomes even worse
since Zerr & Boehler do not include thermal P.

We believe the thermal pressure correction is necessary as discussed in line 240-242. The
pressure increase by 6.6 ~ 9.0 GPa (20 ~ 70 %), depending on temperature, by this correction as
shown in Supplementary Table 1. As is suggested by the referee, if the same correction is
applied to the results of Zerr & Boehler, their melting curve becomes even shallower.

>Almost as an aside, the authors make mention of the controversies in melting temperature in
refractory metal by LHDAC experiments. The possibility that it is occurring due to
"shear-induced anisotropic plastic flow" is an intriguing idea... But shouldn't this happen along
the same P/T path for all samples since it would be dictated by the Re gasket? If this method
proves to be reliable (I'm not yet convinced), this may be good to state it in the discussion, but
not the abstract as it is off topic.

It is clear from our observation that the deformation of the sample was caused by a large
shear stress that is likely induced by the plastic flow of the Re gasket. Whether the deformation
of the gasket occurs during laser-heating or not simply depends on whether the stress increases
beyond the yield strength of Re. Thus, such an extensive gasket/sample deformation is not
commonly observed in other LHDAC experiments heating at moderate temperature (below
~3000 K), but was observed in the present very-high-temperature heating experiments.
However, the gasket deformation may not always result in sample deformation, since the plastic
deformation of the sample also depends on its yield strength. In this sense, temperature plateau

(like the first plateau in this study) caused by sample deformation may be observed at
lower/higher temperature or even not observed depending on materials. Choice of laser (e.g.
CO₂ or YAG) as well as optic system which is directly linked to beam quality such as focused
beam diameter may also influence the behavior.

>Figure 2b, c, d: How much time was the sample heated for between c and d? The temperature
holds steady, but as the gasket got weak, the sample appear to expand. What is causing the
browning of the sample? Did the diamonds burn (especially shown in part d)?

As described above, the heating duration at each laser power was several seconds and the
total heating duration between c and d of the Fig. 2a was less than one minute. Therefore, the
deformation of the sample together with Re gasket occurred gradually over the repeated heating,
where the temperature stayed almost constant with increasing laser power (1st plateau). The
origin of the brown color seen in the laser-heated center is unclear, but might be associated with
the incorporation of Ar pressure medium into the surface of the MgO (particularly, at the grain
boundaries, as shown in the TEM images (Fig. 3c)) during the recrystallization. The presence of
Ar thin layers/droplets at grain boundaries of MgO crystals might cause a scattering of optical
light particularly in a specific wavelength range. A further examination is necessary to clearly
answer to the reviewer's point, but this is definitely not an easy task.

>Figure 3: The agreement with Du et al. (2014) is rather striking, although the estimates of the
melting temperature at the CMB are different by nearly 1000 K (8000 vs 8900 K). Why the
discrepancy?

Extrapolation from our revised melting curve gives a melting temperature of ~7900 K at
the CMB condition, which is in very good agreement with that estimated by Du et al. (2014),
although in general, some uncertainty is always associated with such a large extrapolation.

>Figure S3: There appears to be a slight shift in the MgO peaks between the "HT" and "LT"
regions. What do you attribute this to? Are the MgO (and Ar) volumes consistent for this
pressure? What are the uncertainties? Why are the "HT" peaks broader?

We calculated the pressures at the HT and LT regions from the diffraction patterns using
 the equation of state (EoS) for MgO (Tange et al., 2009) and Ar (Jephcoat, 1998), as shown in
 Table I. The pressure values calculated from the two EoS (for MgO and Ar) are almost
 consistent with each other. The pressure difference between the HT and LT regions is as small
 as ~1 GPa, which can be interpreted to reflect the subtle difference in the stress condition, as
 can be guessed from the very contrasting microtextures between the two regions (line 121-134),
 in addition to the pressure gradient in the sample chamber.

**Table I. Lattice constant of MgO (a_{MgO}) and pressures at the HT and LT regions,**
 **calculated from the equation of state for MgO (P_{MgO}) and Ar (P_{Ar}). ΔP represents the**
 **pressure difference between P_{MgO} and P_{Ar}**

Region	a_{MgO} (Å)	P_{Ar} (GPa)	P_{MgO} (GPa)	ΔP (GPa)
HT	4.05	22.5	23.8	1.3
LT	4.05	23.3	24.7	1.4

>Supplementary Table S2: I'm very puzzled at the relevance on the values given in this table.
 The average over such a large region (20 um x 20 um) is problematic. I'd prefer to see a
 compositional map. Is the melt region enhanced in Ar or the other way around? How
 was Ar quantified? What were the "real" totals before normalization?

Unfortunately, the sample recovered from 32 GPa, 5200K (from which we obtained the
 quantitative data) is not available any more for an additional SEM-EDS analysis, because it was
 used to prepare cross-section foils by FIB for further examination by TEM (STEM-EDS).
 Alternatively, we performed an elemental mapping on one of the cross-section foil by
 STEM-EDS and found many small Ar inclusions within MgO crystals and at the grain
 boundaries, as shown in Supplementary Fig. 4. The origin of the Ar droplets is described above.

**Supplementary Figure 4. Bright-field scanning TEM image and X-ray maps of Mg,**
 **O, Ar, C, and Re collected from the laser-heated area of the sample recovered**
 **from 32 GPa and 5200 K.** Scale bar represents 2 μm . The X-ray maps identified that
 the quenched sample consists of elongated and granular MgO crystals with many small
 Ar inclusions. The concentrations of C and Re are artifacts, derived from the epoxy
 resin and re-deposition from the sputtered Re gasket during Ar ion milling, respectively.

**Minor problems:**

**Line 9: add "most" after, "Periclase (MgO) is the second..."**

We added it in the sentence (line 9).

**Line 10: remove "chemical". This is redundant.**

We removed the word from the sentence (line 10).

Line 11: "mantle-core boundary" should be "core-mantle boundary" to go with standard
convention.

We changed the words in the sentence (line 19).

Line 175: missing "nm" after "500 to 800"

We added it in the sentence (line 214).

Reviewer #1 (Remarks to the Author):

According to the comments raised by reviewers, authors substantially revised the manuscript with additional experimental data and geophysical implications. I'm sure that the manuscript is now more robust and attractive than previous.

However, I'm still not fully convinced by the explanation for lower melting T proposed by Boehler's group. In addition, newly provided T-profile raises one important question, which is not mentioned in the manuscript.

Authors argue that the "first plateau" is due to plastic flow of the sample, Ar medium and gasket, and such deformation can be recognized only at higher T. However, I can hardly see the significant deformation of the (MgFe)SiO₃ sample or gasket in Figure 2 by Zerr and Boehler (1993 Science), although it was heated at around 5000 K with Ar medium and (probably) Re gasket. Is this because of the (MgFe)SiO₃ sample? If so, only MgO is problematic?

Authors newly provided the T-profile during heating (Fig S2). T value are distributed \pm around 500 K as usual in the laser-heated DAC experiments. Which points were used for "temperature" in W-T curve (e.g., Fig 2)? This must be described with reasonable explanation, otherwise the temperature has systematic uncertainty of \pm around 500 K.

Reviewer #2 (Remarks to the Author):

The authors have addressed my comments, and I feel I can recommend the manuscript for publication.

Reviewer #3 (Remarks to the Author):

This is a re-review of this paper and I will first state that this manuscript is much better than the previous version I read. The additional data (e.g., at room P and higher pressures) as well as inclusion of the viscosity was much needed.

The techniques they use, the laser power vs. temperature plateaus and sample texture, are by themselves relatively common (the former more so than the latter), although the way that the authors implement them are novel. Determining that the first laser plateau occurs because of sample/gasket deformation, whereas the following plateau occurs because of melting.

I am conflicted though to recommend publishing in Nature Communications. How about a more specific Earth science journal (Nature Geoscience, GRL, EPSL)?

**Response to the review on our manuscript entitled “Melting**
**temperatures of MgO under high pressure by micro-texture analysis”**

Here are our responses (shown in black) to the reviewer’s comments (in blue).

**Response to Referee #1**

>According to the comments raised by reviewers, authors substantially revised the manuscript
with additional experimental data and geophysical implications. I’m sure that the manuscript is
now more robust and attractive than previous. However, I’m still not fully convinced by the
explanation for lower melting T proposed by Boehler’s group. In addition, newly provided
T-profile raises one important question, which is not mentioned in the manuscript.

>Authors argue that the “first plateau” is due to plastic flow of the sample, Ar medium and
gasket, and such deformation can be recognized only at higher T. However, I can hardly see the
significant deformation of the (Mg,Fe)SiO₃ sample or gasket in Figure 2 by Zerr and Boehler
(1993 Science), although it was heated at around 5000 K with Ar medium and (probably) Re
gasket. Is this because of the (Mg,Fe)SiO₃ sample? If so, only MgO is problematic?

In the melting experiments by Zerr and Boehler (1993) the details of the CO₂ laser optics
(such as the beam diameter on the target and the temperature distribution in the sample under
heating) as well as the detailed information of the gasket material used (Re?) and the sample
chamber (e.g. thicknesses of the sample and Ar pressure transmitting media below and above)
were not mentioned, despite that such information is essential for understanding what actually
occurred during heating (at the temperature plateaus). Therefore, it may not be proper to directly
compare their result on (Mg,Fe)SiO₃ melting with our result on MgO, since the result could
potentially be influenced by such factors. Nevertheless, as the reviewer pointed out, in their
experiment neither a significant deformation of (Mg,Fe)SiO₃ sample nor a plastic flow of the
gasket likely have occurred as far as seeing the images of the sample chamber taken before and
after the laser-heating (Fig. 2A and B of their paper). This may be due to a stress release
associated with the phase transition from enstatite (initial starting material) to perovskite
(bridgmanite) that must have occurred during heating. According to the equations of state of
each phase (Stixrude and Lithgow-Bertelloni, 2005, Geophys. J. Int.) a significant volume
reduction (~10% at 25 GPa and 300K) is expected through the phase transition, and this could

contribute to the relaxation of the thermal stress by laser-heating. Indeed, a signature of the
abrupt volume change can be found in their figure (Fig. 2B of Zerr and Boehler, 1993), where
large cracks (fracturings) were created in the laser-heated spots. Therefore, we suppose the
significant volume change of the sample through the phase transition is a potential reason why
the deformation of the sample/gasket was not observed in Zerr and Boehler's melting
experiment on (Mg,Fe)SiO₃.

>Authors newly provided the T-profile during heating (Fig S2). T value are distributed ± around
500 K as usual in the laser-heated DAC experiments. Which points were used for “temperature”
in W-T curve (e.g., Fig 2)? This must be described with reasonable explanation, otherwise the
temperature has systematic uncertainty of ± around 500 K.

In response to the reviewer's comment, we have added a statement to explain how we
determined the temperatures in the revised text (Line 216-219). The temperature at each laser
power was determined by reading the peak value of the lateral temperature distribution profile,
as shown in the revised figure (Suppl. Fig. 2). The temperature difference from the neighboring
points is as small as ~100 K, which is substantially smaller than the averaged measurement
error of ~200 K.

**Supplementary Figure 2. Typical example of the lateral temperature distribution in the**
**MgO sample under heating at 36 GPa.** The plots and curve represent the temperatures
measured near the hottest spot and the fitting by the Gaussian function, respectively.

Response to Referee #2

>The authors have addressed my comments, and I feel I can recommend the manuscript for
publication.

**Response to Referee #3**

>This is a re-review of this paper and I will first state that this manuscript is much better
than the previous version I read. The additional data (e.g., at room P and higher
pressures) as well as inclusion of the viscosity was much needed.

>The techniques they use, the laser power vs. temperature plateaus and sample texture,
are by themselves relatively common (the former more so than the latter), although the
way that the authors implement them are novel. Determining that the first laser plateau
occurs because of sample/gasket deformation, whereas the following plateau occurs
because of melting.

>I am conflicted though to recommend publishing in Nature Communications. How
about a more specific Earth science journal (Nature Geoscience, GRL, EPSL)?

As discussed in the last paragraph of the main text (Line 183-192), determination of
the melting temperature of refractory metals such as Mo, Ta and W at high pressure by
LHDAC experiments remains controversial, as is the case for MgO. The present study
raises the possibility that shear-induced plastic deformation of the sample (together with
the gasket) during high temperature heating is also responsible for the anomalously
low-temperature melting curves of these metals determined by the earlier LHDAC
experiments. Indeed, a significant anisotropic plastic flow of Ta (and potentially other
refractory metals) before melting is theoretically predicted to cause the underestimation
of the melting temperature determined from the plateau in the power vs temperature
profile by LHDAC (Wu et al., 2009, Nature Material). The present report is probably
the first experimental demonstration of such a novel idea and has profound implications
to understand the real melting behavior of such refractory materials. Therefore, we
believe the present work attracts the interest of a wide range of readers and deserves to
be published in Nature communications.

Reviewer #1 (Remarks to the Author):

I'm satisfied with discussion about (Mg,Fe)SiO₃. I hope it will be incorporated into manuscript. On the contrary, newly explained criterion of temperature looks problematic for me. I can recommend this manuscript for publication in the Nature Communications once authors answer this concern.

Authors newly explained that the "temperature" is taken at the peak of the temperature distribution. I'm strongly concerned that this overestimates the melting temperature. Their microscopic image showed that the melting pool is around 40 μm in diameter (Fig. 3). On the other hand, the temperature is about -500 K / 10 μm lower than peak temperature according to the T-distribution provided in the revised manuscript (Supplementary Fig. 2). At melting temperature, liquid and solid must be coexisting each other, and thus the liquid-solid surface should be at melting temperature. However, the liquid-solid surface is ~20 μm away from the center, where the "melting temperature" was obtained. If I assume T-gradient of -500 K / 10 μm and the diameter of melt pocket = 40 μm, actual melting temperature of MgO can be ~1000 K lower than their result at ~30 GPa.

I believe that the paper is more robust if authors demonstrate the comparison between T-distribution and microscopic image of the recovered sample, which is recently demonstrated by Japanese group using same apparatus (Fig. 4 of Ozawa et al., 2016 EPSL).

**Response to the review on our manuscript entitled “Melting**
**temperatures of MgO under high pressure by micro-texture analysis”**

Here are our responses (shown in black) to the reviewer’s comments (in blue).

**Response to Referee #1**

> I’m satisfied with discussion about (Mg,Fe)SiO₃. I hope it will be incorporated into
manuscript. On the contrary, newly explained criterion of temperature looks problematic for me.
I can recommend this manuscript for publication in the Nature Communications once authors
answer this concern.

We added the discussion on the melting of (Mg,Fe)SiO₃ perovskite by a previous study (Zerr
and Boehler, 1993) as a comparison with the present case for MgO melting in the
Supplementary Notes (line 65-77), according to the reviewer’s request.

> Authors newly explained that the “temperature” is taken at the peak of the temperature
distribution. I’m strongly concerned that this overestimates the melting temperature. Their
microscopic image showed that the melting pool is around 40 μm in diameter (Fig. 3). On the
other hand, the temperature is about -500 K / 10 μm lower than peak temperature according to
the T-distribution provided in the revised manuscript (Supplementary Fig. 2). At melting
temperature, liquid and solid must be coexisting each other, and thus the liquid-solid surface
should be at melting temperature. However, the liquid-solid surface is ~20 μm away from the
center, where the “melting temperature” was obtained. If I assume T-gradient of -500 K / 10 μm
and the diameter of melt pocket = 40 μm, actual melting temperature of MgO can be ~1000 K
lower than their result at ~30 GPa.

> I believe that the paper is more robust if authors demonstrate the comparison between
T-distribution and microscopic image of the recovered sample, which is recently demonstrated
by Japanese group using same apparatus (Fig. 4 of Ozawa et al., 2016 EPSL).

We checked the radial temperature distribution profiles collected during heating at the second
plateau and found that it became flat, while the profiles obtained at lower temperatures showed
a standard Gaussian distribution, as shown in the revised Fig. 2b of our manuscript. Please note
that the temperature-flattened region reached to ~40 μm, which is indeed equivalent to the

lateral dimension of the melting pool observed on the cross-section by SEM (Fig. 3f). Therefore,
we don't think our melting temperatures were overestimated due to the lateral temperature
gradient. The flattening in the profile is most likely due to a rapid increase in heat transfer
caused by the convection of the MgO melt. We added the explanation in the revised manuscript
(line 86-88 and 100-104).